# Alpha-Linolenic Acid and Cardiovascular Events: A Narrative Review

**DOI:** 10.3390/ijms241814319

**Published:** 2023-09-20

**Authors:** Camilla Bertoni, Martina Abodi, Veronica D’Oria, Gregorio P. Milani, Carlo Agostoni, Alessandra Mazzocchi

**Affiliations:** 1Department of Veterinary Sciences for Health, Animal Production and Food Safety, University of Milan, 20122 Milan, Italy; camilla.bertoni@unimi.it (C.B.); martina.abodi@unimi.it (M.A.); 2Pediatric Area, Fondazione IRCCS Ca’ Granda Ospedale Maggiore Policlinico, 20122 Milan, Italy; veronica.doria@policlinico.mi.it (V.D.); gregorio.milani@unimi.it (G.P.M.); 3Department of Clinical Sciences and Community Health, University of Milan, 20122 Milan, Italy; alessandra.mazzocchi@unimi.it

**Keywords:** alpha-linolenic acid, hempseed oil, cardiovascular events, LDL-cholesterol, TC-cholesterol, cardioprotective effect, heart disorders, linoleic acid

## Abstract

Cardiovascular diseases (CVDs) represent the leading cause of global mortality with 1.7 million deaths a year. One of the alternative systems to drug therapy to minimize the risk of CVDs is represented by alpha-linolenic acid (ALA), an essential fatty acid of the omega-3 series, known for its cholesterol-lowering effect. The main purpose of this review is to analyze the effects of ALA and investigate the relevant omega-6/omega-3 ratio in order to maintain functionally beneficial effects. Concerning the lipid-lowering preventive effects, ALA may favorably affect the values of LDL-C and triglycerides in both adult and pediatric populations. Furthermore, ALA has shown protective effects against hypertension, contributing to balancing blood pressure through customary diet. According to the 2009 EFSA statement, dietary ALA may contribute to reducing the risk of CVDs, thanks to anti-hypertensive, anti-atherosclerotic and cardioprotective effects.

## 1. Introduction

Cardiovascular diseases (CVDs) represent the primary cause of global mortality, as they are responsible for about 1.7 million deaths a year, and the major contributor to reduced quality of life [1,2]. 

CVDs include ischemic heart disease (IHD), stroke, heart failure, peripheral arterial disease, and several other cardiac and vascular conditions. As of 2023, the American Heart Association identifies CVDs as the leading cause of death in the United States, accounting for 928,741 deaths in 2020. Between 2018 and 2019, the direct and indirect costs of total CVD were $407.3 billion ($251.4 billion in direct costs and $155.9 billion in lost productivity/mortality) [3].

In addition to conventional pharmacological therapy (ACE inhibitors, beta-blockers, statins, fibrates and PCSK9 inhibitors), nutraceutical solutions have been proposed in recent years as contributing factors to reduce cardiovascular risk [4,5]. Accordingly, nutraceuticals may act on the reduction of lipid risk markers, including total cholesterol (TC), low-density lipoprotein (LDL-C), and triglycerides (TG), and can be divided on the basis of their mechanism of action: sterols and glucomannan may reduce LDL by decreasing the intestinal adsorption of endogenous cholesterol [6,7], while red yeast, garlic, panthetine and policosanols inhibit hepatic cholesterol synthesis [8].

Similarly, plant-derived polyunsaturated fatty acids (PUFAs) may share analogous functional effects. The human body is able to synthesize polyunsaturated fatty acids (PUFAs), except for linoleic acid (LA), the precursor of the omega-6fatty acids, and alpha-linolenic acid (ALA), the precursor of the omega-3 fatty acids. Since these compounds must be introduced preformed with diet, they are called essential fatty acids (EFAs).

Omega-3 and omega-6 fatty acids play a crucial role in brain function, growth, and development as well as in the prevention of heart disease. The unique biochemical structure of ALA is so important because it is involved in certain processes linked to immunity, vision and the production of hormone-like compounds.

Concerning the PUFAs, only ALA and LA are essential fatty acids. They are essential nutrients that cannot be synthesized by the human body and have to be obtained in the diet. Like all fatty acids, ALA and LA are used to provide energy and are stores in adipose tissue; significant amounts are incorporated into cell membranes as well.

ALA and LA are important structural components of cell membranes and are able to influence membrane properties, such as fluidity, flexibility, permeability and activity of membrane-bound enzymes [9,10]. ALA gives origin to longer-chain eicosapenthaenoic acid (EPA) and docosahexaenoic acid (DHA); EPA and DHA are PUFAs, derived from ALA. The main role of EPA is anti-inflammatory: in fact, the cascade of enzymatic reactions to which it is subjected in certain circumstances leads to the production of signal molecules (called “good” eicosanoids) that counteract the pro-inflammatory activity of other similar molecules originating from the so-called arachidonic acid (AA) inflammatory cascade [11,12]. DHA (a 22C compound of the omega-3 series) is a semi-essential fatty acid known for its distinct metabolic activities. Specifically, DHA is attributed with hypolipidemic properties, useful in reducing blood concentrations of triglycerides and LDL; neuroprotective properties, effective in protecting the central nervous system from the damaging effects of reactive oxygen species; antioxidant properties, biologically valuable for various organs and tissues, including the reproductive system; anti-inflammatory properties, capable of shutting down the upstream inflammatory cascade; and immunomodulatory and antiallergic properties [11,12]. DHA is necessary for the development of brain functions and retinal functions associated with vision [13]. Both LA and ALA give origin to eicosanoids from their 20-C derivatives. Eicosanoids are powerful chemical messengers that intervene in the immune and inflammatory response. Increasing the intake of omega-3 fatty acids increases the EPA content of cell membranes, resulting in a higher proportion of eicosanoids derived from EPA. The omega-3- derived eicosanoids may contrast the pro-inflammatory, pro-coagulatory, and atherogenic effects of eicosanoids derived from the omega-6 [9].

Like ALA, LA may have relevant biologic benefits, such as maintaining healthy cell membranes and promoting cell growth. LA is a key nutrient throughout life from birth. In the first months of life, it is supplied to the infant through breast milk, in which this essential PUFA represents approximately 10–15% of total fatty acids [14]. Infant formulas must contain both omega-3 and omega-6 PUFAs for optimal growth and development, with a minimum LA content of 500 mg/100 kcal and a maximum of 1200 mg/100 kcal [15,16], according to Regulation EU 2016/2017 (Supplementing Regulation (EU) Number 609/2013 of the European Parliament and of the Council with regard to “specific compositional and information requirements for infant formulas and follow-on formulas and with regard to infant and young child feeding information requirement”. UE, 2016). The role of derived eicosanoids, within physiologic limits, is beneficial to support reactions against infective agents and clotting processes. Nevertheless, an excess of LA at the expense of ALA may lead to an overproduction of arachidonic acid (AA) [11,12] and derived eicosanoids, representing a risk factor for long-term chronic inflammatory disorders. Also, very high intakes of LA, in the short term, can cause significant metabolic and functional damage through the synthesis of powerful pro-oxidant agents (formation of toxic lipoperoxides, for instance).

LA and ALA compete for the enzyme delta-6-desaturation reaction. This has been suggested to be important for health, as a high intake of LA may reduce the amount of the enzyme available for ALA metabolism, thereby increasing the risk of heart disease through mechanisms mainly associated with pro-inflammatory conditions [17,18,19]. Although ALA is the preferred substrate of the desaturated delta-6 enzyme, an excess of LA in the diet compared to ALA brings more clearly the formation of AA [20]. While the protective effects of the precursor and metabolites of the omega-3 series (ALA and EPA/DHA) have been known for a long time, the role of omega-6, especially LA, in diseases of the cardiovascular system is only recently known [21]. Until recently, LA and its derivatives, AA, were known for pro-inflammatory properties; now, scientific evidence has investigated the role of LA and its benefits in CVD. The balanced intake of LA and ALA is critical in order to give omega-6its anti-inflammatory properties, which is useful in CVDs [22,23]. For this reason, it is important to maintain a proper balance between omega-3 and omega-6 in the diet, because these two substances compete for the enzyme delta-6 desaturase, although they should work together to promote health. Evidence supporting this theory shows that over the past 150 years, Omega-6 intake through today’s diet has increased, while omega-3 intake has decreased in parallel with the increase in heart disease. Thus, the concept of an “ideal” ratio of omega-6/omega-3 in the diet was developed and is between 10:1 and 5:1 [24,25].

Humans evolved on a diet with an omega-6/omega-3 ratio of about 1, while Westerners (e.g., those on Western diets) have a ratio of 15/1–16/1 [20,24]. These diets are deficient in omega-3 fatty acids and contain excessive amounts of omega-6 fatty acids. Excessive amounts of omega-6, including LA, or a very high unbalanced omega-6/omega-3 ratio, promote the pathogenesis of many diseases, including cardiovascular, cancer and inflammatory/autoimmune diseases. In contrast, high levels of ALA and omega-3 exert suppressive effects against these diseases. In the secondary prevention of CVD, a 4/1 ratio was associated with a 70% reduction in total mortality. A ratio of 2.5:1 reduced cell proliferation in colorectal cancer patients, while a lower omega-6/omega-3 ratio was associated with a significant reduction in the risk of breast cancer in women [11,24].

Within this context, while the functional effects of LA are well known [26,27,28,29,30], ALA, present in hempseed oil (HSO), walnuts, and olive and flaxseed oils, still has relevant and functional effects. Nuts and seeds are important sources of ALA and other micronutrients, such as sterols, fibers and polyphenolic compounds. These nutrients are effective in protecting against cardiovascular, inflammatory and chronic diseases [31,32,33]. As for the ALA content of nuts and seeds, 28 g of hempseed or walnuts exceeds the adequate intake for ALA, which is ideally set at 1.1 g/day for women and 1.6 g/day for men [34,35]. The EFSA has set reference values for of omega-3 PUFA intake and, considering cardiovascular health and neurological development, is fixed at about 2–3 g daily intake of ALA, corresponding to 1% of the total energy intake of 18,000/2700 kcal/day [24,36]. Flaxseed, hempseed and canola oil are the main sources of ALA; soybean oil is often considered to be a low-to-moderate source of this nutrient, as studies of its fatty acid composition have shown ALA concentrations ranging from 2.7% to 7.8% [37,38]. These dietary sources of ALA may also be relevant in either pregnant and breastfeeding women, not only for their rich nutritional composition—inclusive of either fat and non-fat compounds vitamins, f.i.—but also because of the well-founded indication to avoid complex mixtures of herbal supplements that may jeopardize the health of both mother and child. In addition, plant sources of omega-3 PUFAs could be considered an effective option for women who cannot tolerate fatty fish and for those who suffer from nausea, a common manifestation during pregnancy [39,40]. While there are numerous studies on the fatty acid composition of human milk, information regarding the types of individual fatty acids provided by the mother to the suckling infant during the first year of life is scarce. Marangoni et al. [41] measured the total fat contents and the concentrations of major fatty acids in pooled breast milk and the following was observed: that total saturated fatty acids gradually increases and the percentage of total monounsaturated fatty acids gradually decreases. Then, it was observed that the amounts of LA and ALA progressively increase. The amounts (mg/kg) of omega-6and omega-3 increase about 2–5 times between colostrum and 1 month, maintaining a very constant ratio, and remain stable up to 3 months [41]. These data suggest that LA is as important as ALA during this period of life in the newborn. LA is critically supportive of infant growth, as are ALA and its series-3 derivatives. A supplementation with ALA to lower the omega-6/omega-3 ratio may lead to up to a 50% reduction in C-reactive protein (CRP), a risk factor for coronary heart disease [9]. Exclusively vegan diets should be evaluated carefully because of the risk of omega-3 PUFA deficiency. In addition to lower intakes of total and saturated fats, another characteristic of exclusively vegan diets is a higher proportional intake of omega-6 PUFAs compared with the more diverse vegetarian diets [42,43]. For these reasons, recommendations for vegan diets to include adequate amounts of ALA are important to keep the advantages of a plant-derived diet [44].

Therefore, the main purpose of this narrative review is to analyze the effects of ALA at the cardiovascular level and to investigate not only the relevant omega-6/omega-3 ratio, but also the absolute amount of ALA in order to maintain functionally beneficial effects [45,46,47].

## 2. Functional Effects of ALA

We selected different studies on ALA and its effects on the cardiovascular system and categorized them by effects on risk factors for CVD and direct cardioprotection.

### 2.1. Risk Factors for Cardiovascular Disease

The most important claim declared by the EFSA concerning ALA is related to the influence on blood cholesterol concentrations [48]. Specifically, the claim indicates that ALA contributes to the maintenance of normal blood cholesterol levels, unlike EPA and DHA, which, being of animal origin, act directly in significantly reducing elevated triglyceride levels. (Figure 1)

A cause-effect relationship between the dietary intake of ALA and the reduction of plasma concentrations of TC and LDL-C has been recognized. Studies issued after the EFSA statement further suggest that ALA has anti-proliferative [49], anti-hypertensive [50,51,52], anti-atherosclerotic [53,54] and cardioprotective effects [53,54,55,56,57,58], and it successfully improves the composition of the red cell membrane in children with hyperlipidemia [19,59], possibly reducing the risk of cardiovascular events in adulthood. Similar to ALA, there are numerous studies that have focused on the use of the precursor of the omega-6 series, LA, in CVDs. A 2020 study by Marangoni et al. [30] clarifies the correlation between LA intake and the risk of CVDs. This narrative review states that high tissue and blood LA concentrations are associated with lower cardiovascular risk and reduced mortality. In fact, dietary guidelines suggest that high consumption of omega-6, along with omega-3, reduced CVD risk, although more studies with longer follow-up are needed to clarify the role of LA in this field. The 2018 study by Hooper et al. [60] also confirms this aspect: significant intakes of omega-6, particularly LA, have been associated with reduced risk of heart attack, lowered TC-C, and decreased risk of not only cardiac but also cerebral events. The efficacy of this nutrient at reducing TG levels and LDL-C, increasing HDL-C and reducing BMI remains unclear. Therefore, from a cardiovascular standpoint, both ALA and LA may be effective. Based on this, a 2023 study [61] concluded that intake of trans fatty acids (palmitic acid, stearic acid, and saturated fatty acids of animal origin) was modestly associated with a higher risk of mortality and CHD, whereas intake of ALA, long-chain omega-3 fatty acids, and LA was modestly associated with a lower risk. Supplementation with long-chain omega-3 fatty acids and increased consumption of ALA and LA instead of saturated fats reduced the risk of coronary events. Taken together, the available evidence provides modest support for current recommendations to replace saturated fats with PUFAs [60].

Although it has been said that LA and omega-6 are also effective in treating CVDs, it is known that they play a prominent role in CVD risk, sometimes increasing the risk of CVD. In fact, the beneficial effects of LA are dose-dependent: excess LA is considered potentially harmful to the body because it induces high production of AA, which appears to alter the body’s inflammatory balance. High levels of LA induce AA and the production of bad eicosanoids with pro-inflammatory activity. Increased inflammation is a major factor that greatly increases CVD risk. Thus, excess LA is detrimental to omega-3 metabolism. Since omega-6 as LA is found in many more foods than ALA, it is easy to understand how easy it is to overdo LA intake and AA production [23].

In the study conducted by Greupner et al. [19], the aim was to compare two extreme LA to ALA ratios involving a low LA/high ALA diet with a ratio of 0.5–1:1 and a high LA/low ALA diet with an LA ratio of 20–30:1 on fatty acid concentrations in red blood cells (RBCs), with emphasis on EPA and DHA. To obtain the two LA/ALA ratios in the diet, the subject’s fatty acid intake was tightly controlled by a multistep method. During the course of the low-LA/high-ALA diet, ALA concentrations increased rapidly (*p* < 0.001) from 1.44 ± 0.17 μg·mL^−1^ at baseline to 5.63 ± 0.45 μg·mL^−1^ at day 7 and to 6.34 ± 0.63 μg·mL^−1^ at day 14, corresponding to a mean change of 332 ± 40% and 354 ± 47%, respectively, whereas in the high-LA/low-ALA diet, ALA concentrations decreased after 7 days from 1.47 ± 0.13 μg·mL^−1^ to 1.09 ± 0.09 μg·mL^−1^ (*p* = 0.011) and increased again on day 14 to 1.41 ± 0.17 μg·mL^−1^, which is not significantly different from the baseline level. The results showed that the low-LA/high-ALA diet is effective in increasing ALA and EPA concentrations in red blood cell membranes. The findings that EPA concentrations in red blood cells increased significantly by 35.0 ± 13% at 7 days and by 57.6 ± 18% at 14 days after the low-LA/high-ALA diet is likely a result of increased conversion of ALA to EPA, since no EPA was ingested with the basal diet.

In a pilot study by Del Bo’ et al. [59], an 8-week-long randomized clinical trial (RCT) dietary intervention study aimed to evaluate the impact of HSO supplementation on the lipid profile and fatty acid composition of RBCs in children and adolescents with primary hyperlipidemia. The 36 study subjects, aged 6 to 16 years, were divided into two different groups: the control group and the HSO group, receiving 3 g of HSO with 1.4 g of linoleic acid (LA) and 0.7 g/day of ALA. Both groups received specific dietary guidelines. Blood samples were kept for each subject, before and after administration with HSO, in order to analyze the lipid profile, composition of RBCs, and omega-3 index. After an eight-week supplementation with HSO, there were significant reductions in the RBC content of total saturated and monounsaturated FAs (−5.02 ± 7.94% and −2.12 ± 2.23%, respectively). Conversely, the levels of total omega-3 and omega-6 PUFAs increased (+1.57 ± 1.96% and +5.39 ± 7.18%, respectively), as did the omega-3 index (+1.18 ± 1.42%). This study confirms that diet represents the first line of therapy for primary hyperlipidemia.

In the clinical study by Yue et al. of 2021 [62], the effect of ALA intake on blood lipid profiles was examined, especially on triglycerides (TG), TC, HDL-C, LDL-C, VLDL-C and the ratio of TC to HDL-C. A total of 1305 subjects were enrolled in the ALA group and 1325 were enrolled in the control group. Compared with the control group, dietary intake of ALA significantly reduced the concentrations of TG (WMD −0.101 mmol/L; 95% CI: −0.158 to −0.044 mmol/L; *p*  =  0.001), TC (WMD −0.140 mmol/L; 95% CI: −0.224 to −0.056 mmol/L; *p*  =  0.001), LDL-C (WMD −0.131 mmol/L; 95% CI: −0.191 to −0.071 mmol/L; *p*  <  0.001), VLDL-C (WMD −0.121 mmol/L; 95% CI: −0.170 to −0.073 mmol/L; *p*  <  0.001), TC/HDL-C ratio WMD −0.165 mmol/L; 95% CI: −0.317 to −0.013 mmol/L; *p*  =  0.033) and LDL-C/HDL-C ratio (WMD −0.158 mmol/L; 95% CI: −0.291 to −0.025 mmol/L; *p*  =  0.02). ALA has no effect on HDL-C (WMD 0.008 mmol/L; 95% CI: −0.018 to 0.034 mmol/L; *p*  =  0.541). Dose-response analysis showed that a 1 g/day increase in ALA was associated with 0.0016 mmol/L, 0.0071 mmol/L, 0.0015 and 0.0061 mmol/L reductions in TG (95% CI: −0.0029 to −0.0002 mmol/L), TC (95% CI: −0.0085 to −0.0058 mmol/L), HDL-C (95% CI: −0.0020 to −0.0011 mmol/L) and LDL-C (95% CI: −0.0073 to −0.0049 mmol/L) levels, respectively. The effects of ALA intake on TG, TC, and LDL-C concentrations were pronounced in patients with hyperlipidemia or hyperglycemia compared with healthy subjects. Dietary ALA intervention improved blood lipid profiles by reducing levels of TG, TC, LDL-C and VLDL-C. These results show that increasing ALA intake could potentially prevent the risk of CVDs.

In a meta-analysis of controlled trials conducted by Khalesi et al. [63], the intake of ALA-rich sesame fractions is associated with a reduction in TG. As a result, the consumption of sesame did not significantly change the TC (−0.32 mmol/L, 95% CI: −0.75 to 0.11; *p* = 0.14, I2 = 96%), LDL-C (−0.15 mmol/L, 95% CI: −0.50 to 0.19; *p* = 0.39, I2 = 96%) or HDL-C levels (0.01 mmol/L, 95% CI:−0.00 to 0.02; *p* = 0.16, I2 = 0%). However, a significant reduction was observed in serum TG levels (−0.24 mmol/L, 95% CI: −0.32 to −0.15; *p* < 0.001, I2 = 84%) after consumption of sesame. Although the consumption of sesame rich in ALA seems to significantly reduce blood TG levels. 

The positive effects of ALA can also be extended to non-lipid outcomes as well. Experiments show that ALA stimulates nitric oxide (NO) production [64,65], and increases action mediated by prostanoids, with effects on platelet aggregation and coagulation. Therefore, it reduces the probability of thrombotic events, regulates the heart rhythm and decreases the onset of arrhythmias and inflammation, with a direct action on prostaglandins [66]. Recent evidence agrees that the intake of ALA has a cardioprotective effect, through mild lowering of blood pressure (BP). Dietary ALA intake was reported to be epidemiologically associated with a lower prevalence of hypertension, and ALA has been indicated as a promising alternative addition to available lifestyle medications for the prevention of CVD [52]. The present study was designed to comparatively investigate the effects of ALA and LA supplementation against hypertension and the underlying molecular mechanisms. Hypertensive rats and normotensive control rats were used. They were given an LA diet (10% canola oil, 20% protein, 54% carbohydrate and 26% fat) and an ALA-supplemented diet (10% flaxseed oil, 20% protein, 54% carbohydrate and 26% fat). The animals were randomly divided into four groups with different diets for 8 weeks. After 8 weeks of dietary treatment, ALA supplementation significantly reduced systolic blood pressure, whereas LA supplementation showed no apparent effects. In addition, ALA supplementation not only reduced BP but also improved endothelial function. In conclusion, the data showed that ALA could be used in dietary interventions for the prevention and treatment of hypertension [50].

A series of studies aimed to understand how the intake of ALA in the diet was useful in tracing BP and in increasing the aortic thickness of the intima and media tunica in small-gestational-age (SGA) infants [50,51]. In a 2015 study by Skilton et al. [51], 1009 participants were recruited at 6 months and were followed until age 19 years. The purpose of this study was to evaluate a possible association between dietary ALA intake, low BP and aortic intima-media thickness in children born with SGA. BP and food records were assessed at each visit. A total of 1009 participants had at least one blood pressure measurement and complete birth weight and gestational age data, including 115 (11%) with SGA. These children had higher systolic BP and pulse pressure (PP) from age 14 years onwards. In SGA infants, systolic blood pressure (SBP) was 2.1 mmHg lower ([95% CI 0.8–3.3]; *p* = 0.001) and PP 1.4 mmHg lower ([95% CI 0.3–2.4]; *p* = 0.01), per exponential increase in ALA intake. It can be concluded from the study that ALA supplementation during childhood improves cardiovascular health of children with SGA.

Flaxseed and HSO contain omega-3 fatty acids in the fatty components, with lignans, and fiber as non-fat components, and may be beneficial for patients with CVDs. Hypertension is often associated with peripheral artery disease. Rodriguez-Leyva and colleagues [52] aimed to investigate the effects of daily flaxseed intake on SBP and diastolic blood pressure (DBP) in patients with peripheral artery disease. In this prospective, double-blinded, placebo-controlled, randomized trial, 110 patients consumed a variety of food containing 30 g of ALA or placebo daily for 6 months. Plasma levels of omega-3 ALA increased 2- to 50-fold in the flaxseed-fed group but did not increase significantly in the placebo group. SBP was ≈10 mmHg lower, and DBP was ≈7 mmHg lower in the flaxseed group compared with placebo at 6 months. Enrolled patients with an SBP ≥ 140 mmHg at baseline achieved a significant reduction of 15 mmHg in SBP and 7 mmHg in DBP with flaxseed supplementation. These results confirmed that ALA is one of the most powerful antihypertensive agents achieved by a dietary intervention.

The MARGARIN trial [67] examined changes in atherosclerosis markers in 110 subjects with a high risk of IHD. The experimental group received margarine (80% fat, of which, 60% as PUFAs) containing either 15% or 0.3% of total fat as ALA for two years. Results showed that the intake of ALA reduced C-reactive protein, a marker of inflammation, but the present study found no effect on markers of atherosclerosis.

### 2.2. Cardioprotective Effect

The 2020–2025 American Dietary Guidelines showed that there is strong evidence to demonstrate that replacing saturated fatty acids (SFAs) with PUFAs reduces the risk of IHD events and CVD mortality [64]. In high-risk patients, ALA prevents coronary heart disease (CHD), considered one of the major causes of death worldwide [52,53,54,55,57,68,69,70,71,72,73,74,75,76].

The aim of the study by Vedtofte et al. [69] was to examine the association between ALA intake and the risk of CHD. Data from eight American and European prospective cohort studies including 148,675 women and 80,368 men were used. During 4–10 years of follow-up, 4,493 CHD events and 1,751 CHD deaths occurred. Among men, the researchers found an inverse association between ALA intake and CHD events and deaths. Each additional gram of ALA was associated with a 15% lower risk of CHD events (HR: 0.85; 95% CI: 0.72, 1.01) and a 23% lower risk of CHD deaths (HR: 0.77; 95% CI: 0.58, 1.01). The Cardiovascular Health Study [31] was designed to examine the associations of dietary ALA with the risk of mortality, CHD and stroke among older adults who participated in the study, a cohort study of 2709 adults aged ≥ 65 years. After adjustment for age, sex, race, enrollment site, education, smoking status, diabetes, body mass index (BMI), alcohol consumption, treated hypertension and total energy intake, higher dietary ALA intake was found to be associated with a lower risk of total and non-cardiovascular mortality. When the highest quintile of dietary ALA was compared with the lowest quintile, the HR for total and non-cardiovascular mortality was found to be 0.73 (95% CI: 0.61, 0.88) and 0.64 (95% CI: 0.52, 0.80), respectively. In conclusion, this study suggests that dietary ALA is associated with a lower risk of total and non-cardiovascular mortality in older adults.

In a 2012 meta-analysis of observational studies by Pan et al. [53], increasing dietary ALA was associated with a moderately lower risk of total CVD (RR: 0.90; 95% CI: 0.81, 0.99). Twenty-seven studies were analyzed, including 251,049 individuals and 15,327 CVD events. The association between ALA intake and reduced risk of CVD was significant in 13 comparisons, but 17 comparisons showed similar but non-significant trends. Therefore, considering observational studies, higher ALA intake is associated with a moderately lower risk of CVD.

In the study by Zelniker et al. of 2021 [77], after intake of omega-3 PUFAs, patients with acute coronary syndrome experienced a lower risk of cardiovascular death (OR: 0.82; 95% CI: 0.68, 0.98 per 1-SD increment), and an attenuated relation was observed after ALA administration, specifically (OR: 0.92; 95% CI: 0.74, 1.14).

In the European Prospective Investigation into Cancer and Nutrition (EPIC) study [78] which included 22,043 subjects in Greece, there were 275 deaths after 44 months. These individuals consumed a traditional Mediterranean diet rich in ALA, and there were significant reductions in both total mortality and death from CHD of 25% and 33%, respectively. In 1302 individuals with known CHD, there were reductions in total mortality and CHD of 27% and 31%, respectively [79].

The Lyon Diet Heart Study [80] was an effective study in demonstrating the efficacy of a Mediterranean-style diet supplemented with ALA on composite measures of IHD recurrence after a first myocardial infarction (MI). Subjects in the experimental group were instructed to follow a Mediterranean diet and were given a canola-oil-based margarine containing 4.8% ALA. After 46 months of this diet, these subjects had a 50–70% lower risk of recurrent IHD [81]. Only ALA was significantly associated with improved prognosis (RR for the composite of cardiac death and nonfatal acute MI: 0.20; 95% CI: 0.05, 0.84) when plasma fatty acids were analyzed as crude estimates of dietary data.

Another systematic review and meta-analysis of cohort studies by Jingkai et al. [70] examined the overall association between ALA intake and CHD risk, assessing dose-response relationships. Fourteen studies of 13 cohorts were identified and included in the meta-analysis. The pooled results showed that higher ALA intake was associated with a modestly reduced risk of combined CHD (risk ratios (RR) = 0.91; 95% CI: 0.85, 0.97) and fatal CHD (RR = 0.85; 95% CI: 0.75, 0.96). Compared with individuals with lower ALA intake, only subjects with ALA intake <1.4 g/d showed a reduced risk of composite CHD. ALA intake was linearly associated with fatal CHD, and each 1 g/d increase in ALA intake was associated with a 12% reduction in the risk of fatal CHD (95% CI: −0.21, −0.04).

To better understand the role of ALA, Sala-Vila et al. [82] prospectively evaluated the association between dietary ALA and fatal CVD in participants of the PREvèncion with DIeta MEDiterranea (PREDIMED) study (*n* = 7202). These results showed that dietary ALA at > 0.7% of daily energy intake was associated with a 28% reduced risk of all-cause mortality. Participants (*n* = 7447) at high CVD risk in a treatment arm receiving 30 g/day of mixed nuts (15 g walnuts, 7.5 g hazelnuts, and 7.5 g almonds) had a reduced incidence of cardiovascular events (HR: 0.72; 95% IC: 0.54, 0.95) with 4.8 daily grams of ALA consumption compared to the control group. In the intervention group, dietary ALA increased by 0.43 g/day and plasma ALA increased by 0.30%–0.44% in a random subsample [79]. In addition, several studies have examined the association between ALA intake and CHD risk and evaluated a possible dose-response relationship. Results showed that higher intakes of ALA were associated with a modestly reduced risk of composite coronary disease (RR = 0.91; CI 95% 0.85, 0.97) and fatal coronary disease (RR = 0.85; 95%; CI 0.75, 0.96). Subjects who consumed ALA < 1.4 g/day had a significant risk reduction, in contrast to those who did not include ALA in their diet [70].

Regarding ALA and its association with cardiovascular risk, a working hypothesis suggests that ALA may influence cardiovascular risk through effects on arrhythmogenesis and lethal ventricular arrhythmia. Intravenous infusions of ALA reduced the risk of ventricular fibrillation during coronary artery ischemia in different animals, and dietary ALA is associated with a reduced risk of abnormal repolarization in men and women [83]. In addition, after intake, ALA can be converted to EPA and DHA, both of which have antiarrhythmic effects. Other plausible pathways by which ALA may exert beneficial effects on coronary and CVD risk include endothelial function, inflammation and thrombosis. With respect to CVD, only one large trial has evaluated ALA supplementation for cardiovascular outcomes, including arrhythmias. In the AlphaOmega trial [84], ALA was associated with a significant reduction in arrhythmia-related events compared with placebo or EPA and DHA in post hoc analyses in the subgroup of patients with diabetes, who are particularly prone to ventricular arrhythmias and sudden death after MI (HR: 0.39; 95% CI: 0.17, 0.88). In addition, the results of the study by Kromhout et al. [85] suggest an antiarrhythmic effect of ALA intake, but further clinical trials are needed to confirm this. Meta-analyses of observational studies have shown that increasing dietary ALA is associated with a 10% lower risk of total CVD and a 20% lower risk of fatal CHD.

Dietary intake of ALA reduces atherogenesis and inhibits arterial thrombosis [54]. ALA exerts beneficial cardiovascular effects on atherosclerosis and inflammation and has been shown to induce antithrombotic and antiplatelet effects and to reduce mortality [71,72,73,74,75,76]. Dietary ALA was incorporated into adipose tissue and shifted the balance toward the anti-inflammatory class of the thromboxane/prostacyclin mediators [68]. In the study by Stivala et al. [54], eight-week-old male mice were fed a high-fat diet containing 7.3 g of ALA or a low-fat diet containing 0.03 g of ALA for 16 weeks. ALA was administered as flaxseed oil, while cocoa butter was used in the control group. The results showed that dietary ALA increased the platelet count to a biologically relevant extent (51%) without affecting bleeding times. This finding is very relevant and gives the possibility to consider ALA as a powerful and effective element potentially used to reduce thrombotic risk and, consequently, reduce the risk of CVD.

## 3. Discussion

This narrative review describes the functional effects of ALA in preventing risk factors for CVDs, and its direct cardioprotective effects.

Regarding the lipid-lowering preventive effects, this review shows that ALA may favorably affect the levels of LDL-C and TGs in both adult and pediatric populations [59,60,61,62,63]. Dietary ALA intake has a significant effect on the plasma values of LDL-C, TC, and TG levels, as well as on the improvement of erythrocyte membrane composition. However, the reduction of cholesterol and triglyceride levels may also depend on the plant source of ALA used. For example, HSO allows for a significant reduction of LDL-C, TC and RBC, while sesame seeds are effective in reducing triglyceride levels, but not other cholesterol-rich fractions.

A beneficial and protective effect against hypertension has been reported in three studies [50,51,52]. All of them confirm that daily consumption of ALA reduces SBP and DBP, possibly through the release of NO, a potent vasodilator. The results were also confirmed in subjects with borderline BP values, suggesting that ALA is an effective dietary antihypertensive compound.

With regard to cardioprotective effects, the studies and meta-analyses [52,53,54,55,57,69,70,71,72,73,74,75,76,77,78,79,80,81,82,83], reported by us, show controversial results. While some studies confirm an important association between daily ALA intake and a significant reduction in CVD risk, others indicate that there is no strong correlation. A dose-dependent dietary intake of ALA appears to be associated with a reduced risk of CVD. Some studies confirm that adherence to the Mediterranean diet, with the addition of ALA following the correct intakes, has positive and beneficial effects on significantly reducing IHD, cardiac death and nonfatal acute MI. Although some studies are significant and vouch for the effectiveness of ALA on CVDs, further clinical trials are needed to confirm the relationship between ALA intake at different doses and CVD events. Assuming more ALA, for example, by eating more hempseed or flaxseed oil, probably makes little or no difference in all-cause or cardiovascular death or coronary events, but slightly reduces CVD, coronary mortality and heart arrhythmias. The effects of ALA on stroke are unclear, because the evidence was of low quality.

Three studies [83,84,85] investigated the association between ALA intake and its antiarrhythmic effect, caused, probably, by its direct action on anti-inflammatory eicosanoids. In fact, due to the action of these metabolites, the regulation of heart rhythm is also affected. Some of these trials verified that ALA intake was associated with a significant reduction in arrythmia, and subsequently MI, while others need further clinical evidence to confirm the efficacy.

ALA has the potential to provide novel and promising research perspectives, which are indirectly related to CVD prevention. A growing body of evidence [86,87,88,89,90,91,92] suggests that ALA intake may be a co-adjuvant intervention to modulate the progression of inflammatory and cancer-related conditions. The effects of omega-3 PUFAs on cancer stem-like cells (CLSCs) may be an important target for cancer therapy and will be an interesting challenge for future studies. In any case, the antitumor activity of omega-3 PUFAs, shown through multiple mechanisms, suggests that they may have an important therapeutic role in the management of cancer stem cells (CSCs). At this stage, further large observational and prospective studies are needed to confirm these effective and innovative properties of ALA and to develop RCTs.

## 4. Conclusions

Based on the studies and research presented herein, increasing ALA in the daily diet within the recommended omega-6/omega-3 ratio is safe. Thus, in addition to serving as an essential fatty acid and a precursor to more bioactive long-chain omega-3 fatty acid derivatives, ALA has its own significant functional properties similar to those associated with nutraceuticals. As stated in the 2009 EFSA report, the inclusion of dietary ALA may play a role in reducing the risk of cardiovascular events (CHD and IHD) due to its anti-hypertensive, anti-atherosclerotic, and cardioprotective effects. Given the paucity of data in the literature regarding the use of ALA in the setting of CVD, this narrative review aims to summarize the main effects not so much of the metabolites produced by omega-3 (EPA and DHA), but of ALA alone. From the reported trial and meta-analyses, it is still not well understood whether it is more useful and effective to dose the omega-6/omega-3 ratio or to evaluate only the absolute amount of ALA for the prevention of CVDs. More studies are needed to investigate this further. Recently, there has been promising evidence of its potential anti-inflammatory and anti-proliferative properties, which may extend beyond CVDs to include inflammatory and oncological conditions. However, further validation through high-quality studies is required to definitively establish these effects.

## Figures and Tables

**Figure 1 ijms-24-14319-f001:**
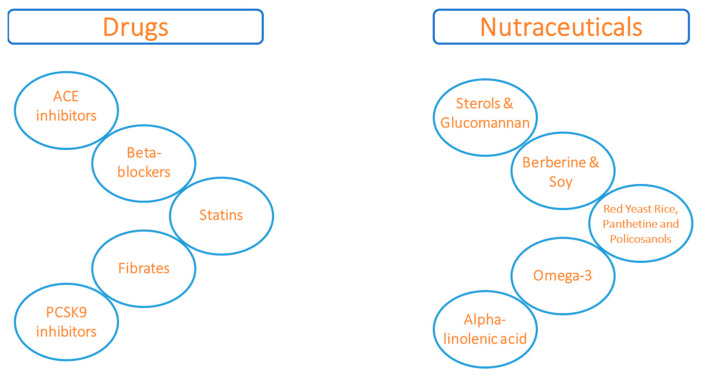
Drugs and nutraceuticals for treating heart failure and lipid risk markers.

## Data Availability

Data sharing not applicable.

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
