# Peer review of "Alpha-Linolenic Acid and Cardiovascular Events: A Narrative Review"

_ijms, 2023, doi:10.3390/ijms241814319_

Round 1

Reviewer 1 Report

The manuscript submitted for review addresses the interesting issue of the role of alpha-linolenic acid in cardiovascular disease.

However, the manuscript should be subject to major revision before possible publication. My main complaint is that the treatment of the topic is too cursory, the work superficially presents the effects of ALA and only mentions the ratio of omega 6 to omega 3 fatty acids, and this was the goal according to the authors.

The autors listed numerous references, and only sparse of them are discussed.

Other comments are as follows:

lines 39-53 - the fragment may be removed

line 65 - "less known effects" - in fact, ALA is studied quite widely

lines 92-94 - triglycerides seem to be at least equally important considering ALA effects

Figure 1 - unclear, is it jutst a list of drugs and nutraceuticals? Does the order matter?

Sesame and hempseed oil are not the richest sources of ALA

lines 142-152 - PUFAs contain not only ALA

minor errors, especially punctuation, should be corrected

Author Response

We thank you for your suggestions and comments. We have to tried to apply them in order to make the manuscript clearer.

However, the manuscript should be subject to major revision before possible publication. My main complaint is that the treatment of the topic is too cursory, the work superficially presents the effects of ALA and only mentions the ratio of omega 6 to omega 3 fatty acids, and this was the goal according to the authors. Regarding the effects of ALA: we have covered all aspects affecting the cardiovascular area (atherosclerosis, anti-inflammatory, anti-hypertensive, and cholesterol-lowering activity) (lines 140-309). The main objective of our manuscript is to investigate not only the n-6/n-3 ratio, but also to identify the absolute amount of ALA that may be useful to administer as a form of prevention in the context of CVD. “Therefore, the main purpose of this narrative review is to analyze the effects of ALA at the cardiovascular level and to investigate not only the relevant n-6/n-3 ratio, but also the absolute amount of ALA in order to maintain functionally beneficial effects [38,39,40].” (ref. lines 115-117)

“From  the reported trial and meta-analyses, it is still not well understood whether it is more useful and effective to dose the n-6/n-3 ratio or to evaluate only the absolute amount of ALA for the prevention of CVDs. More studies are needed to investigate this further.” (ref. lines 351-354). In the conclusion, we again marked the choice to focus on the main objective, which is not only to evaluate the ALA/LA ratio, but to dose absolute amounts of ALA to see which of the two values is more predictive in cardiovascular prevention.

The authors listed numerous references, and only sparse of them are discussed. We discussed all studies and meta-analyses concerning the effects of ALA at the CVD level.

Other comments are as follows:

lines 39-53 - the fragment may be removed: We removed  the fragment (line 39-53) because we think we regressed too long and too detailed about alternative nutraceuticals to plant-based omega 3, the topic on which our review focuses. We have summarized alternative nutraceutical solutions in a few lines (line 41-43).

line 65 - "less known effects" - in fact, ALA is studied quite widely. At line 95, we modified the text: “ALA, present in hempseed oil (HSO), walnuts, olive and flaxseed oils, has still relevant and functional  effects”

lines 92-94 - triglycerides seem to be at least equally important considering ALA effects. Regarding the effect on triglycerides, ALA is not directly indicated in treatment. On triglyceride levels, omega 3 of animal origin, the metabolites, such as EPA and DHA act effectively. (line 126-129)

Figure 1 - unclear, is it jutst a list of drugs and nutraceuticals? Does the order matter. The figure 1 is only a graphic representation of the therapeutic solutions indicated for the treatment of LDL-C hypercholesterolemia. The order is not important.

Sesame and hempseed oil are not the richest sources of ALA. The richest plant sources of ALA are walnuts, nuts, flax and hemp oil and seeds.

lines 142-152 - PUFAs contain not only ALA. We have replaced the study by Cicero et al with a more recent one (2020), concerning not PUFAs in general, but one that discussed how ALA compared with LA, is an excellent way to reduce blood pressure. (lines 202-210)

Reviewer 2 Report

This study titled “Alpha-linolenic acid and cardiovascular events: a narrative review” discussed the effect of ALA on lipid profile and cardiovascular diseases. It could provide some nutraceutical guidance of ALA to the daily diet choice to better benefit health and reduce the risk of cardiovascular-related diseases. My major concerns are:

1. What is the novelty of this review? How it is different from other similar reviews?

2. The cited references are a little out of date. The number of references published in 5 years is less than 40%. Please try to cite newer references to make the review more up-to-date.

3. Line 49 and 50- what is the purpose of mentioning Berberine here?

Minor comments:

1.       Please use an international unified format to present the number symbol. For example, a. Line 28- what is “1,7 million death”? 1.7 million? b. Line 181- “148.675 women and 80.368 men” should be “148,675 women and 80,368 men”.

2.       About the reference numbering in the context, for the consecutive numbers, please do not list them one by one, e.g., in line 256, “[41,42,43,45,55,56,57,58,59,60,61,62,63,64]” could be simply as “[41-43, 45, 55-64]”. Please check all the reference numbers.

Author Response

We thank you for your suggestions and comments. We have to tried to apply them in order to make the manuscript clearer.

  1. What is the novelty of this review? How it is different from other similar reviews? We think that the novelty is that we have focused only on the heart benefits of not all Omega 3, but those of plant origin. We are not talking about EPA and DHA, but in depth about ALA, a long-chain polyunsaturated fatty acid only of plant origin, a precursor of Omega 3 series.. It would also be an update review regarding ALA-CVD correlation, as data in the literature regarding ALA are very scarce (ref. in the conclusions. Lines 349-351)
  2. The cited references are a little out of date. The number of references published in 5 years is less than 40%. Please try to cite newer references to make the review more up-to-date. Unfortunately, the scientific literature regarding the cardiovascular benefits of ALA are really outdated. We have tried to select relatively recent ones:

- Mejia-Zepeda R, Perez-Hernández IH. Effect of alpha linolenic acid on membrane fluidity and respiration of liver mitochondria in normoglycemic and diabetic Wistar rats. J Bioenerg Biomembr. 2020;52(6):421-430

- Stivala S, Reiner MF, Lohmann C, Lüscher TF, Matter CM, Beer JH. Dietary α-linolenic acid increases the platelet count in ApoE−/− mice by reducing clearance. Blood. 2013;122(6):1026-33

- Mozaffarian D, Wu JHY. Omega-3 fatty acids and cardiovascular disease: effects on risk factors, molecular pathways, and clinical events. J Am Coll Cardiol. 2011 (20);2047-2067

- Eussen SRBM, Geleijnse JM, Giltay EJ, Rompelberg CJ, Klungel OH, Kromhout D. Effects of n-3 fatty acids on major cardiovascular events in statin users and non-users with a history of myocardial infarction. Eur Heart J. 2012(13);1582-1588

- De Caterina R. N-3 fatty acids in cardiovascular disease. N Engl J Med. 2011(364):2439-2450

- Winnik D, Lohmann C, Richter EK. Dietary α-linolenic acid diminishes experimental atherogenesis and restricts T cell-driven inflammation. Eur Heart J. 2011(32):2573-2584

- Li G, Wang X, Yang H, Zhang P, Wu F, Li Y, Zhou Y, Zhang X, Ma H, Zhang W, Li J. α-Linolenic acid but not linolenic acid protects against hypertension: critical role of SIRT3 and autophagic flux. Cell Death Dis. 2020;11(2):83

- Djuricic I, Calder PC. Beneficial Outcomes of Omega-6 and Omega-3 Polyunsaturated Fatty Acids on Human Health: An Update for 2021. Nutrients. 2021;13(7):2421

- Elagizi A, Lavie CJ, O’Keefe E, Marshall K, O’Keefe JH and Milani RV. An Update on Omega-3 Polyunsaturated Fatty Acids and Cardiovascular Health. Nutrients. 2021;13(1):204

  1. Line 49 and 50- what is the purpose of mentioning Berberine here? We have removed the reference to berberine, because we believe we havegone too far in discussing nutraceutical alternatives to ALA that share the same mechanism of action.

Minor comments:

  1. Please use an international unified format to present the number symbol. For example, a. Line 28- what is “1,7 million death”? 1.7 million? b. Line 181- “148.675 women and 80.368 men” should be “148,675 women and 80,368 men”. We uniformed 1.7 million at 28th line and 148.675 women and 80.368 men at 240th line.
  2. About the reference numbering in the context, for the consecutive numbers, please do not list them one by one, e.g., in line 256, “[41,42,43,45,55,56,57,58,59,60,61,62,63,64]” could be simply as “[41-43, 45, 55-64]”. Please check all the reference numbers. We modified all the reference numbering:

- line 95: [20-24]

- line 142: [46-51]

- line 238: [45-48, 50, 61-68]

- line 313: [52-55]

- line 323: [45-48, 50, 61-76]

- line 338: [80-86]

Reviewer 3 Report

In his review the authors summarise the potential use of ALA in CVD. 

1. Metabolism ALA need to be given.

2. Daily intake o ALA and how it is metabolized and what are products formed and their actions need to be discussed.

3. Why and how ALA is better than LA in CVD.

4. What is the cell membrane composition of LA and ALA.

5. Any studies that looked at ALA content of leukocytes, platelets, macrophages and endothelial cells and their relationship to CVD.

6. Any studies on the intake of ALA and CVD-clinical trials and their results.

7. Why ALA need to be considered as better than LA in the prevention of CVD.

8. In what ay ALA is different from LA in its role in CVD.

9. Why a balance between LA and ALA need to be maintained. 

ok

Author Response

We thank you for your suggestions and comments. We have to tried to apply them in order to make the manuscript clearer.

  1. Metabolism ALA need to be given. All references regarding metabolism of ALA are from 51 to 71 and from 78 to 82 lines.
  2. Daily intake o ALA and how it is metabolized and what are products formed and their actions need to be discussed. All references about daily intake are from 99 to 102 lines. About the products of ALA (EPA+DHA) all references are from 55 to 71 lines.
  3. Why and how ALA is better than LA in CVD. From 145 to 151 references about differences between ALA and LA in CVDs.
  4. What is the cell membrane composition of LA and ALA. All references are from 54 to 55 lines, and then from 152 to 173 lines (studies by Greupner and Del Bo’).
  5. Any studies that looked at ALA content of leukocytes, platelets, macrophages and endothelial cells and their relationship to CVD.

- lines 152-163: in this study of Greupner et al, the authors focused on the ability of ALA to improved endothelial function in order to treat hypertension.

- lines 303-309: Stivala et al in this study showed that dietary ALA increased platelet count to a biologically relevant extent without affecting bleeding times. This is very relevant and gives the possibility to consider ALA as powerful and effective element used to reduce thrombotic risk.

  1. Any studies on the intake of ALA and CVD-clinical trials and their results.

- lines 152-163: study by Greupner about ALA intake and modification in RBCs

- lines 164-173: study by Del Bo’ about ALA intake and the possibility to use this as therapy for primary hyperlipidemia

- lines 174-188: study by Yue et al about ALA intake and his effect on hypercholesterolemia and reduction of CVD risk

- lines 198-210: study about relationship between ALA intake and his hypertensive effect

- lines 211-220: study by Skilton et al about ALA intake, improvement of cardiovascular health in children wit SGA and reduction of BP

- lines 223-230: study by Rodriguez-Leyva et al about the effects of daily flaxseed intake on SBP and DBP in patients with peripheral artery disease

- lines 239-251: study by Vedtofte et al examines the association between ALA intake and the risk of CHD

- lines 252-256: meta-analysis by Pan et al studies intake of ALA and a moderately lower risk of total CVD

- lines 264-269: Lyon Diet Heart Study focused on the relationship between ALA and improvement of prognosis about CVD risk

- lines 270-276: Jinkay et al study an overall association between ALA intake and CHD risk. ALA was linearly associated with fatal CHD

- lines 277-287: Sala-Vila et al in PREDIMED study focuses on the daily intake of ALA and the reduction risk of all-cause mortality (CHD, fatal coronary disease)

- lines 295-300: AlphaOmega trial studied how ALA intake was associated with a significant reduction in arrythmia-related events, compared with placebo or EPA/DHA

- lines 304-309: Stivala et al in this study showed that dietary ALA increased platelet count to a biologically relevant extent without affecting bleeding times. This is very relevant and gives the possibility to consider ALA as powerful and effective element used to reduce thrombotic risk.

  1. Why ALA need to be considered as better than LA in the prevention of CVD. From 145 to 151 references about differences between ALA and LA in CVDs.
  2. In what ay ALA is different from LA in its role in CVD. From 145 to 151 about differences between ALA and LA in CVDs.
  3. Why a balance between LA and ALA need to be maintained. Lines 82-94.

Round 2

Reviewer 1 Report

I am satisfied with the authors' explanations and the changes they made to the text of the manuscript.

Author Response

We thank you for your comments. 

Reviewer 2 Report

The authors have addressed all of my questions. 

Author Response

We thank you for your comments. 

Reviewer 3 Report

Despite elaborate explanations offered by the authors to my previous questions, it is still debatable whether LA is better or ALA is better in preventing CVD. 

There are many studies that sowed that LA is important to prevent CVD.

see the following papers:

1. Vergroesen AJ. Dietary fat and cardiovascular disease: possible modes of action of linoleic acid. Proc Nutr Soc. 1972 Dec;31(3):323-9. doi: 10.1079/pns19720059. PMID: 4579353.

2. Jayedi A, Soltani S, Emadi A, Ghods K, Shab-Bidar S. Dietary intake, biomarkers and supplementation of fatty acids and risk of coronary events: a systematic review and dose-response meta-analysis of randomized controlled trials and prospective observational studies. Crit Rev Food Sci Nutr. 2023 Aug 26:1-20. doi: 10.1080/10408398.2023.2251583. Epub ahead of print. PMID: 37632423.

3. SOLOFF LA, SCHWARTZ H. ERYTHROCYTIC PHOSPHOLIPID LINOLEIC ACID IN ISCHAEMIC HEART-DISEASE AND IN HEALTH. Lancet. 1964 Dec 12;2(7372):1268-9. doi: 10.1016/s0140-6736(64)92738-2. PMID: 14219130.

4. MOORE JH, WILLIAMS D. THE RELATIONSHIP BETWEEN THE LINOLEIC ACID CONTENT OF THE DIET, THE FATTY ACID COMPOSITION OF THE PLASMA PHOSPHOLIPIDS AND THE DEGREE OF AORTIC ATHEROSIS IN EXPERIMENTAL RABBITS. Br J Nutr. 1964;18:603-12. doi: 10.1079/bjn19640054. PMID: 14241591.

5. ABITBOL L, VILLEA AA, PEREIRA AF. [Clinical experiences with linoleic acid and pyridoxine in hypercholesterolemia]. Rev Bras Med. 1960 Apr;17:322-4. Portuguese. PMID: 13681041.

6. Insull W Jr, Lang PD, Hsi BP. Adipose tissue fatty acid differences in American men between 1962 and 1966. Am J Clin Nutr. 1970 Jan;23(1):17-26. doi: 10.1093/ajcn/23.1.17. PMID: 4244116.

7. Kirkeby K, Nitter-Hauge S, Bjerkedal I. Fatty acid composition of adipose tissue in male Norwegians with myocardial infarction. Acta Med Scand. 1972 Apr;191(4):321-4. PMID: 5032676.

8. Kalstad AA, Tveit S, Myhre PL, Laake K, Opstad TB, Tveit A, Schmidt EB, Solheim S, Arnesen H, Seljeflot I. Leukocyte telomere length and serum polyunsaturated fatty acids, dietary habits, cardiovascular risk factors and features of myocardial infarction in elderly patients. BMC Geriatr. 2019 Dec 27;19(1):376. doi: 10.1186/s12877-019-1383-9. PMID: 31881852; PMCID: PMC6935134.

9. Hooper L, Al-Khudairy L, Abdelhamid AS, Rees K, Brainard JS, Brown TJ, Ajabnoor SM, O'Brien AT, Winstanley LE, Donaldson DH, Song F, Deane KH. Omega-6 fats for the primary and secondary prevention of cardiovascular disease. Cochrane Database Syst Rev. 2018 Nov 29;11(11):CD011094. doi: 10.1002/14651858.CD011094.pub4. PMID: 30488422; PMCID: PMC6516799.

10. Hooper L, Al-Khudairy L, Abdelhamid AS, Rees K, Brainard JS, Brown TJ, Ajabnoor SM, O'Brien AT, Winstanley LE, Donaldson DH, Song F, Deane KH. Omega-6 fats for the primary and secondary prevention of cardiovascular disease. Cochrane Database Syst Rev. 2018 Jul 18;7(7):CD011094. doi: 10.1002/14651858.CD011094.pub3. Update in: Cochrane Database Syst Rev. 2018 Nov 29;11:CD011094. PMID: 30019765; PMCID: PMC6513455.

11. Sanders TA. Protective effects of dietary PUFA against chronic disease: evidence from epidemiological studies and intervention trials. Proc Nutr Soc. 2014 Feb;73(1):73-9. doi: 10.1017/S0029665113003789. Epub 2013 Dec 5. PMID: 24308351.

12. Mayneris-Perxachs J, Guerendiain M, Castellote AI, Estruch R, Covas MI, Fitó M, Salas-Salvadó J, Martínez-González MA, Aros F, Lamuela-Raventós RM, López-Sabater MC; for PREDIMED Study Investigators. Plasma fatty acid composition, estimated desaturase activities, and their relation with the metabolic syndrome in a population at high risk of cardiovascular disease. Clin Nutr. 2014 Feb;33(1):90-7. doi: 10.1016/j.clnu.2013.03.001. Epub 2013 Mar 28. PMID: 23591154.

13. Czernichow S, Thomas D, Bruckert E. n-6 Fatty acids and cardiovascular health: a review of the evidence for dietary intake recommendations. Br J Nutr. 2010 Sep;104(6):788-96. doi: 10.1017/S0007114510002096. Epub 2010 Jun 4. PMID: 20522273.

14. Shen J, Zhang L, Wang Y, Chen Z, Ma J, Fang X, Das UN, Yao K. Beneficial Actions of Essential Fatty Acids in Streptozotocin-Induced Type 1 Diabetes Mellitus. Front Nutr. 2022 May 19;9:890277. doi: 10.3389/fnut.2022.890277. PMID: 35669071; PMCID: PMC9164285.

15. Das UN. The lipids that matter from infant nutrition to insulin resistance. Prostaglandins Leukot Essent Fatty Acids. 2002 Jul;67(1):1-12. doi: 10.1054/plef.2002.0374. PMID: 12213429.

I recommend the authors to present a more balanced view of LA and ALA in CVD rathe than giving a biased opinion that is favoring ALA. 

I may add here that both LA and ALA are present in human breast milk LA > ALA) implying that LA is more important for human physiology compared to ALA. 

nil

Author Response

We thank you for your suggestions and comments and for giving us the opportunity to improve the manuscript and make it clearer, trying to have dealt with the parallel roles of ALA and LA in the best possible way. 

There are many studies that sowed that LA is important to prevent CVD. We thank you for providing many references regarding the effect of LA on cardiovascular prevention. We selected 2 of those presented (refs. 60-61), as well as the most recent and the most relevant to the subject of our review. In addition (ref. 30) we considered the 2020 study by Marangoni et al.

In lines 86 to 91, we covered how, not only ALA, but also the role of LA and Omega 6 is important in improving cardiovascular health (refs. 21-22-23). In lines 73-76, we emphasized the role of ALA, stressing its essentiality in a person’s life, considering that breast milk is particularly rich in it, in excellent balance with the precursor and derivatives of the Omega 3 series. The presence of LA in breast milk is essential to ensure proper growth and development of the central nervous system in the newborn, in full compliance with maximum and minimum values (refs. 14-15-16). Precursor of Omega 6, LA, represents approximately 10-15% of total fatty acids in breast milk.

In lines 163-172, we analyzed the articles by Marangoni (ref. 30), Hooper (ref. 60) and Jayedi (ref. 61), who argue that higher intake of LA and Omega 6 are correlated with improved cardiovascular health. Marangoni et al (ref. 30) confirm the correlation between LA high intake and reduced mortality. Jayedi et al (ref. 60)  conducted a study to assess effects of increasing Omega 6 fats (LA) on CVD and all-cause mortality. The authors found evidence that increased intake of Omega 6 fats may make little or no difference to all-cause mortality or CVD events. Then, high-quality evidence suggests increasing Omega 6 fats reduces TC-C parameters, reduces the risk of heart attack, and reduces the risk of cardiovascular and cerebral events. LA intake, however, does not guarantee a reduction in LDL-C, triglycerides, BMI, and an increase in HDL-C. These values are improved by an ALA, rather than LA intake. Studies are still needed to verify and ensure the effectiveness of Omega 6 on these types of hematochemical parameters. It remains certain that in order or reduce mortality and the risk of acute CHD, it is necessary to replace saturated fats with PUFAs (ref. 60).

I recommend the authors to present a more balanced view of LA and ALA in CVD rather than giving a biased opinion that is favoring ALA.

I may add here that both LA and ALA are present in human breast milk LA > ALA) implying that LA is more important for human physiology compared to ALA. In lines 73-76, we emphasized the role of ALA, stressing of its essentiality in a person’s life, considering that breast milk is particularly rich in it, in excellent balance with the precursor and derivatives of the Omega 3 series (refs. 14-15-16). The presence of LA in breast milk is essential to ensure proper growth and development of the central nervous system in the newborn, in full compliance with maximum and minimum values. Precursor of Omega 6, LA, represents approximately 10-15% of total fatty acids in breast milk. In lines 118-126 (ref. 40),  the study by Marangoni et al were measured the total fat contents and the concentration of major fatty acids in pooled breast milk. Data suggest that total saturated fatty acids gradually increase and the total MUFAs gradually decrease. The amount of ALA and LA progressively increase and this suggest that LA and ALA are very important for this period of life in the newborn, supporting  infant growth.

Round 3

Reviewer 3 Report

nil

nil